# Few-bit Backward: Quantized Gradients of Activation Functions for Memory Footprint Reduction

## Abstract

Memory footprint is one of the main limiting factors for large neural network training. In backpropagation, one needs to store the input to each operation in the computational graph. Every modern neural network model has quite a few pointwise nonlinearities in its architecture, and such operation induces additional memory costs which — as we show – can be significantly reduced by quantization of the gradients. We propose a systematic approach to compute optimal quantization of the retained gradients of the pointwise nonlinear functions with only a few bits per each element. We show that such approximation can be achieved by computing an optimal piecewise-constant approximation of the derivative of the activation function, which can be done by dynamic programming. The drop-in replacements are implemented for all popular nonlinearities and can be used in any existing pipeline. We confirm the memory reduction and the same convergence on several open benchmarks.

## 1 Introduction

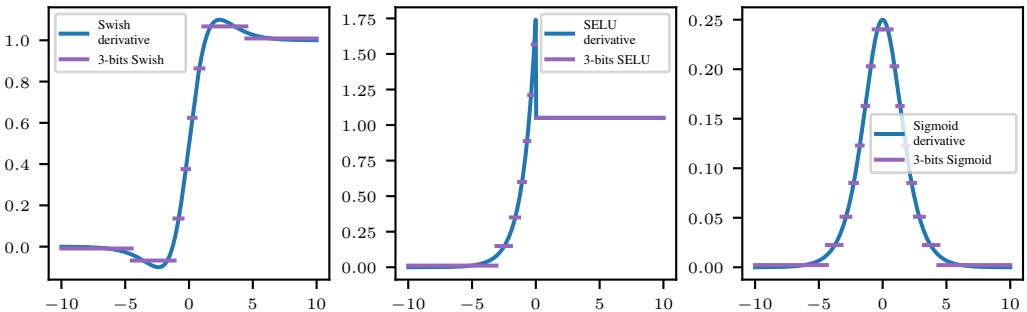

Figure 1: Examples of 3-bit approximations for derivatives of popular nonlinearities: GELU, SELU, and Sigmoid.

Modern neural network models are getting larger and larger. One of the main bottlenecks in the training loop is the required device memory storage Ojika et al. (2020); Gao et al. (2020). In this paper, we propose a universal approach that helps to reduce the model memory footprint during backpropagation. Note that this approach is complementary to other memory reducing techniques such as checkpointing Chen et al. (2016) or offloading Beaumont et al. (2021). Our method can be applied to any neural network without any additional preprocessing.

Memory consumed by the model during training (except intermediate tensors) can be split into two groups: 1) the model weights (including additional memory for the optimizer state), 2) activations saved for the backward pass, over which the computation is not carried out directly at the moment, but which will be required in the future to compute the gradients.

Every operation in the computational graph generates a memory footprint. It is typically overlooked, that the application of the pointwise non-linearity (such as GELU or sigmoid) results in storing the

input for the backward pass. We show that instead of keeping the full input tensor, it is possible to store a low-bit representation, which allows accurate gradients approximation.

In this work, we propose to approximate the derivative of the activation function in a piecewise-constant form. Such an approximation problem has to be solved once for each activation function, and we propose a simple technique to do that.

The proposed approximation divides all values into several bins and saves only their corresponding bin indices instead of storing all values. This is a lossy compression, but the additional noise introduced by it is negligible as we will show on several benchmarks in Section 4.

The main contributions of our paper are:

- We propose new approximate backward computation schemes that significantly reduce the memory consumption of neural network training.

- We benchmark our approach on several tasks. We show that it provides up to 40% memory reduction on various tasks while maintaining accuracy on par with the model trained via the standard approach.

## 2 QUANTIZED GRADIENTS OF ACTIVATIONS

**Gradients of activations using automatic differentiation.** Modern deep learning frameworks use the *reverse mode automatic differentiation* to calculate the gradients of the loss over the model parameters. Forward computation can be associated with a directed acyclic graph, depicted in Fig. 2. Each operation $f$ computes the output $\mathbf{X}_{l+1}$ given the input $\mathbf{X}_l$ and has to save some information $\mathbf{S}_l$ that would be used on the backward pass in order to calculate the derivative $\partial L/\partial \mathbf{X}_l$ from $\partial L/\partial \mathbf{X}_{l+1}$ and $\mathbf{S}_l$. Thus, in a typical training loop, the intermediates $\mathbf{S}_l$ of all operations in the graph are stored in the memory during the whole forward pass until they are no longer needed after the completion of the corresponding backward operation during backward pass. This generates an additional memory, which can be quite significant and be larger than the total amount of parameters of the model.

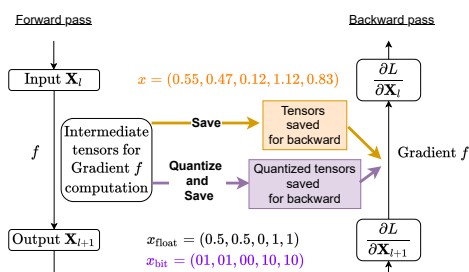

Figure 2: Computation graph of both forward and backward pass. Orange and purple parts of the graph correspond to standard and proposed ways of saving tensors for backward, respectively. Vector $x_{\text{bit}}$ stands for the tensor saved using 2-bit quantization, while $x$ denotes its uncompressed version.

**Pointwise activations.** In this paper, we focus on a pointwise activation function, which is ubiquitous in modern neural network architectures. Given an input tensor $\mathbf{X}_l$ we apply a function $f$ to each of the elements of this tensor:

$$f(\mathbf{X}_l) = [f(\mathbf{X}_l^{j_1,\ldots,j_k})]_{j_1,\ldots,j_k}, f : \mathbb{R} \to \mathbb{R}.$$

This operation is very cheap compared to other operations in the deep neural network model and does not attract much attention when analysing computational complexity. However, standard implementation in such a framework as PyTorch induces not a very small memory footprint and the whole input $\mathbf{X}_l$ is saved for the backward pass.

The backward pass for such a function consists of element-wise multiplication of the propagated gradient tensor by the derivative of the nonlinearity function at the points of the input tensor: if $\mathbf{X}_{l+1} = f(\mathbf{X}_l)$, then the gradient of the loss $L$ with respect to $\mathbf{X}_l$ is computed as

$$\frac{\partial L}{\partial \mathbf{X}_l} = \frac{\partial L}{\partial \mathbf{X}_{l+1}} f'(\mathbf{X}_l), \tag{1}$$

where $f'(\mathbf{X}_l)$ is the tensor with elements, consisting of the derivative of $f$ evaluated in each element of $\mathbf{X}_l$. From Eq. (1), it follows that for the backward pass we have to store only $f'(\mathbf{X}_l)$, and $\mathbf{X}_l$ is not needed.

**ReLU activation function.** To illustrate our idea, consider one of the most popular nonlinearities, $f(x) = \text{ReLU}(x) = \max(0, x)$. Its derivative $f'$ takes only two values, 0 and 1 and it only requires 1 bit to store. If single precision is used, then the compression is 32, which is quite noticeable.

**GELU activation function.** In modern transformer architectures Vaswani et al. (2017) the GELU Hendrycks & Gimpel (2016) nonlinearity is typically used. The derivative no longer takes two values. Instead, we propose to approximate $f'$ by a *piecewise-constant* function. For example, if we allow 8 different values, we will need only 3 bits per each element (Fig. 1).

**Quantized gradients of activations.** In stochastic optimization, if the gradient for a given batch is computed approximately, the optimization may still converge. The GELU derivative (see Fig. 1) is quite "similar" to a piecewise-constant function: for large values of $|x|$, it is almost exactly equal to 0 or 1, and for small values of $x$, a rather interesting transition from 0 to 1 occurs. Instead of calculating the derivative exactly on the backward pass, we approximate it using a certain piecewise-constant approximation:

$$q(x|\mathbf{s}, \mathbf{y}) = \sum_{i=1}^{k} y_i \mathbb{1}[x \in [s_i; s_{i+1}]], \quad (2)$$

where $\mathbf{s} = (s_1, \cdots, s_{k+1})$ is a sorted vector of intervals, on which approximation is constant, $\mathbf{y} = (y_1, \cdots, y_k)$ is a vector of the correspond-

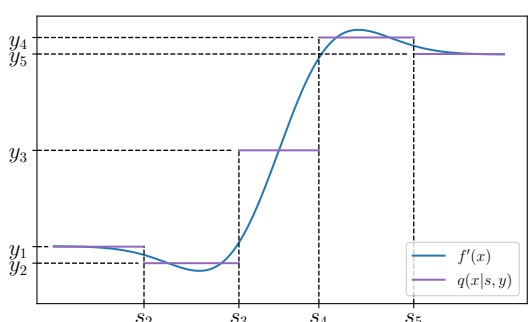

Figure 3: GELU derivative and its approximation $q(x|\mathbf{s}, \mathbf{y})$ with five piecewise-constant intervals

ing values of approximation and $\mathbb{1}$ denotes an indicator function, which equals 1 whenever its argument is true and 0 otherwise. That means, that $q(x|\mathbf{s}, \mathbf{y})$ equals $y_i$ when $x \in [s_i; s_{i+1}]$, see Fig. 3 for illustration. As noted above, if the approximation has $k$ constant intervals, instead of storing the full input tensor $X$, it will be possible to save only $\log k$ bits of information (per element of the input tensor), which, accordingly, will reduce the memory consumption by $32/\log k$ times for single precision.

If quantizatoin scheme Eq. (2) is given, drop-in replacement for activation function $f$ is very straightforward. On the forward pass, instead of the full tensor $\mathbf{X}$, we have to save only indices of intervals to which the elements of $\mathbf{X}$ belong, and on the backward pass, we need to multiply gradient w.r.t. output not with the actual derivative of $f$, but with values from $\mathbf{y}$ corresponding to stored indices. Pseudocode is presented in Alg. 1.

```
1   # Globally stored piecewise-constant approximation parameters
2   s, y = [...], [...]
3
4   def forward(X):
5       X_pos = sortedsearch(s, X)
6       save_for_backward(X_pos)
7       return f(X)
8
9   def backward(dLdY):
10      X_pos = get_saved_for_backward()
11      return dLdY * y[X_pos]
```

Listing 1: Pseudo code for quantized backward layer. Arrays $\mathbf{s}$ and $\mathbf{y}$ are parameters of quantization Eq. (2), sortedsearch is a binary search method.

**Memory of Few-bit Appproximation.** As it was mentioned above, by replacing all pointwise nonlinearity layers in the neural network with Few-bit approximation consisting of $k$ piecewise-constant intervals, the memory consumption of such layers during forward-backward pass will be

reduced by $32/k$ times for single-precision learning mode. However, how many times in total the neural network memory consumption is reduced depends on the particular architecture of the neural network and the optimizer used in the process. During training, the memory is spent on weights (parameters) of the network, on optimizer statistics, and on all stored activations, some of which are activations of pointwise nonlinearity layers. For example, when training ResNet18 with the Adam optimizer on 256x256 images, the model weights take 44.6Mb, $3 * 44.6 = 133.8$Mb is used by the optimizer to store gradients and moments, $BS * 40$Mb is needed to store all activations during forward-pass, $BS * 11.5$Mb of which are pointwise nonlinearity layers and $BS * 28.5$Mb is for all other layers, where $BS$ is the batch size. Therefore, the maximum possible batch size with standard nonlinearities is $\lfloor (\text{GPU\_MEM} - 4 * 44.6)/40 \rfloor$, while the maximum batch size with Few-bit nonlinearities of size $k$ is $\lfloor (\text{GPU\_MEM} - 4 * 44.6)/(28.5 + 11.5 * \log k/32) \rfloor$, where GPU\_MEM is the available GPU memory. In our example with ResNet18 for standard nonlinearity layers, the maximum batch size for a video card with 32Gb memory is 813, while using 4-bit Few-bit approximation is 1086 (+33%). Memory consumption for different Few-bit mods and different neural network architectures is presented in Appendix B.

**Speed of Few-bit Approximation** The memory gain of a Few-bit layer does not slow down the speed. The standard nonlinearity layer calculates the activation function in the forward pass and the activation function gradient in the reverse pass. The activation function gradient usually includes complex functions such as exponent, erf, and others. The Few-bit version of the layer also calculates the activation function on forward pass, but the gradient calculation during backward pass is replaced by one binary search and one lookup in the value table (see Alg. 1). Our efficient implementation of this procedure using CUDA kernels runs several percent faster than the standard nonlinearity layer. However, this result may depend on specific framework implementation and the used GPU, so in our experiments in Section 4 we do not consider the time gain, assuming that both layers are roughly equally fast, but focus specifically on memory savings.

## 3 OPTIMAL PIECEWISE-CONSTANT APPROXIMATION

Fig. 1 shows examples of an optimized 3-bit piecewise-constant approximation for several nonlinearity function. Finding the optimal approximation parameters (boundaries of intervals and values on them) is a challenging task. We propose to find them by minimizing the (weighted) $L_2$ norm of the error.

Consider function $f : \mathbb{R} \rightarrow \mathbb{R}$ and its derivative $f'$. We will measure the quality of a piecewise constant approximation Eq. (2) with a weighted $L_2$ norm:

$$\min_{\mathbf{y},\mathbf{s}} L(\mathbf{s}, \mathbf{y}), \tag{3}$$

$$L(\mathbf{s},\mathbf{y}) = \int_{\mathbb{R}} (f'(x) - q(x|\mathbf{s},\mathbf{y}))^2 w(x) dx = \sum_{i=1}^{k} \int_{s_i}^{s_{i+1}} (f'(x) - y_i)^2 w(x) dx, \tag{4}$$

where $w$ is some weight function reflecting our prior knowledge of the activation function argument distribution. Practical choices of $w$ may be either $\mathbb{1}[x \in [A; B]]$ (with some reasonable $A$ and $B$, which should be large enough) which makes integral Eq. (3) tractable, or maybe, e.g., standard normal distribution.

$L(\mathbf{s}, \mathbf{y})$ is differentiable w.r.t. $\mathbf{s}$ and $\mathbf{y}$, so optimal piecewise-constant approximations can be found using standard gradient-based optimization techniques. But the minimization problem Eq. (3) has many local minima that are far from optimal. We suggest using *dynamic programming* to get some good initial approximation that can be further finetuned using gradient-based methods (but also can be used as is because it is very accurate on its own).

**Dynamic programming.** We will assume that the weighting function $w$ is chosen such that $w(x) = 0$ for $x \notin [A; B]$. Consider the following auxiliary value:

$$\text{DP}(t, k) = \min_{\substack{y_{1:k}, \\ s_{1:k+1}, \\ s.t. s_1 = A, s_{k+1} = t}} \int_{A}^{t} (f'(x) - q(x|\mathbf{y},\mathbf{s}))^2 w(x) dx,$$

$$t \in \mathbb{R}, k \in \mathbb{N}.$$

Essentially, $\mathrm{DP}(t,k)$ is the optimal piecewise constant approximation of size $k$ for the given function $f'$ on the interval $[A;t]$. The recurrent formula for this value is:

$$\mathrm{DP}(t, k+1) = \min_{t'} \left\{ \mathrm{DP}(t', k) + \int_{t'}^{t} (f'(x) - y(t', t))^2 w(x) dx \right\}, \tag{5}$$

$$y(t', t) = \frac{\int_{t'}^{t} w(x) f'(x) dx}{\int_{t'}^{t} w(x) dx}, \tag{6}$$

since a piecewise-constant approximation of size $k+1$ consists of corresponding approximation of size $k$ (first term) plus one constant interval (second term). Here $t'$ chooses the right bound of approximation of size $k$, and $y(t', t)$ stands for the optimal value for the interval $[t'; t]$ Eq. (8). Then the minimal value of $L(\mathbf{s}, \mathbf{y})$ of size $k$ is equal to $\mathrm{DP}(B, k)$.

To solve the minimization problem Eq. (5), we suggest considering the discretization of $t$: $A = t_0 < t_1 < \cdots < t_n = B$ and reducing the calculation of $\mathrm{DP}(t, k)$ to its approximation only in the points of discretization:

$$\mathrm{DP}(i, k) = \min_{j} \left\{ \mathrm{DP}(j, k-1) + T(j, i) \right\},$$

$$T(j, i) = \int_{t_j}^{t_i} (f'(x) - y(j, i))^2 w(x) dx, \quad y(j, i) = \frac{\int_{t_j}^{t_i} w(x) f'(x) dx}{\int_{t_j}^{t_i} w(x) dx}. \tag{7}$$

Eq. (7) can be calculated in $\mathcal{O}(n^2 K)$ time and $\mathcal{O}(nK)$ space, which is described in Appendix G in detail. Please note, that this routine should be evaluated only once, possibly by the framework developers, and the used indefinitely. Which means that number of discritization points $n$ can be taken quite large, tens of thousands easily. That would make global solutoin of discrete problem, given in Eq. (7) very close to the global solution of the original problem Eq. (3). We give precalculated Few-bit approximations for many different pointwise nonlinearity functions in our implementation at `https://github.com/anonymous/repository`.

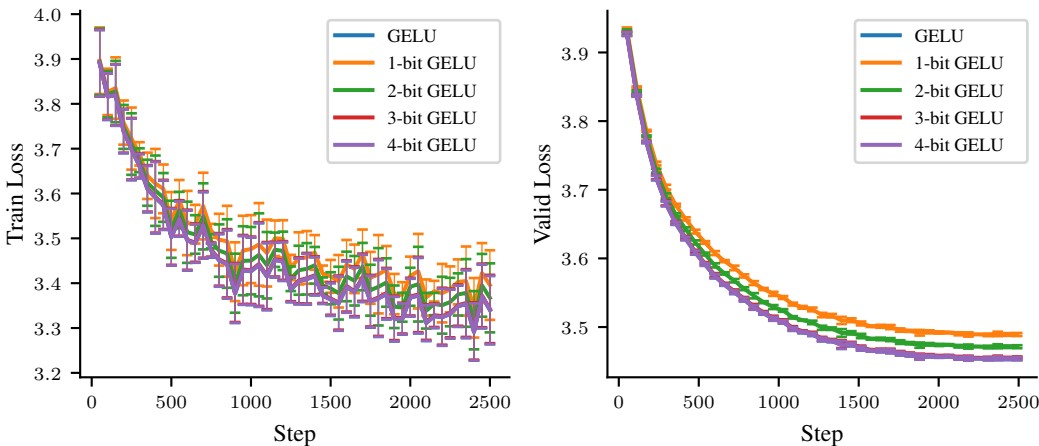

Figure 4: Dynamic of loss values in finetuning of ruDALL-E Malevich with Few-bit GELU activations.

## 4 EXPERIMENTS

The goal of our experiments is not only to show that the Few-bit nonlinearity approach provides memory savings during neural network training without loss of the final model quality. In addition, we want to experimentally prove that this approach does not change the learning dynamic itself because, in this case its application in practice is almost completely safe: there is a memory gain without loss of speed or quality, and without risks of interference with other training factors under study (hence, no additional search or fitting of other hyperparameters is needed). To achieve this

goal, in addition to the main metrics of the trained model (which depend on specific tasks and benchmarks), we also compare the training loss and validation loss graphs during the whole training process. Further you will see that 1-bit and 2-bit f-bit approximations are already almost the same as the original nonlinearity layers. And the 3- and 4-bit Few-bit approximations achieve the original quality of the model.

We have tested two of the most important and commonly used neural network architectures: convolutional neural networks and transformer-based networks. We use standard popular open-source benchmarks with open hyperparameters for training in order to demonstrate the behavior of the Few-bit approach under drop-in replacement of standard nonlinearities without any hyperparameter optimization or specially selected training conditions. In Section 4.1, we test the RoBERT-a transformer-based neural network on the GLUE Wang et al. (2019) benchmark, which includes 9 different NLP tasks. In Section 4.2, we test the training of the generative ruDALL-e model in the task of modeling the joint distribution of text and image tokens for the Russian Emoji dataset. We use the GELU nonlinearity for both transformer architectures, as it is the main nonlinearity function used in such models. In Section 4.3, we test the classical ResNet18 architecture on the ImageNet dataset using the open benchmark ffcv Leclerc et al. (2022). In the classical ResNet architecture, we replace all ReLU nonlinearities with one of GELU, SELU, or Swish to demonstrate that the Few-bit approach works with a wide range of different popular activation functions.

The main analog of our Few-bit approach is the ActNN method. In Section 4.4, we make a detailed comparison with this method.

The code to reproduce all experiments is available at `https://github.com/anonymous/repository`, and all hyperparameters for training are presented in Appendix F.

**4.1 GLUE benchmark.**  In Table 1 we report results for RoBERTa-base model Liu et al. (2019) on GLUE benchmark Wang et al. (2019) for standard GELU and 1-, 2-, 3- and 4-bits Few-bit GELU. 1- and 2-bits versions have minor performance degradation, while 3- and 4-bits GELU have no visible difference and closely match vanilla GELU performance, which can be seen more clearly on the dependence of the metric, averaged across all GLUE tasks, on the number of bits in Few-bit approximation, represented in Fig. 6. The behaviour of loss during training is depicted in Fig. 5: 3- and 4-bit versions are hardly distinguishable from the standard GELU.

Table 1: RoBERTa-base on GLUE benchmark with different quantization budgets. Metric: mean accuracy/correlation (task specific). Averaged across five runs.

|  | 1-bit GELU | 2-bits GELU | 3-bits GELU | 4-bits GELU | Vanila GELU |
|---|---|---|---|---|---|
| stsb | 0.906 ($\pm$ 0.002) | 0.907 ($\pm$ 0.002) | 0.910 ($\pm$ 0.002) | 0.909 ($\pm$ 0.002) | 0.909 ($\pm$ 0.001) |
| mnli-mm | 0.870 ($\pm$ 0.001) | 0.870 ($\pm$ 0.002) | 0.871 ($\pm$ 0.002) | 0.870 ($\pm$ 0.001) | 0.871 ($\pm$ 0.002) |
| mrpc | 0.880 ($\pm$ 0.009) | 0.884 ($\pm$ 0.008) | 0.884 ($\pm$ 0.007) | 0.885 ($\pm$ 0.008) | 0.882 ($\pm$ 0.005) |
| cola | 0.595 ($\pm$ 0.016) | 0.580 ($\pm$ 0.014) | 0.596 ($\pm$ 0.015) | 0.607 ($\pm$ 0.014) | 0.604 ($\pm$ 0.013) |
| mnli | 0.873 ($\pm$ 0.001) | 0.872 ($\pm$ 0.002) | 0.874 ($\pm$ 0.001) | 0.874 ($\pm$ 0.002) | 0.874 ($\pm$ 0.001) |
| sst2 | 0.939 ($\pm$ 0.003) | 0.938 ($\pm$ 0.003) | 0.941 ($\pm$ 0.004) | 0.941 ($\pm$ 0.003) | 0.943 ($\pm$ 0.002) |
| rte | 0.752 ($\pm$ 0.021) | 0.756 ($\pm$ 0.023) | 0.780 ($\pm$ 0.014) | 0.771 ($\pm$ 0.025) | 0.771 ($\pm$ 0.017) |
| qqp | 0.914 ($\pm$ 0.001) | 0.915 ($\pm$ 0.000) | 0.916 ($\pm$ 0.001) | 0.916 ($\pm$ 0.001) | 0.916 ($\pm$ 0.001) |
| qnli | 0.925 ($\pm$ 0.002) | 0.925 ($\pm$ 0.002) | 0.926 ($\pm$ 0.002) | 0.927 ($\pm$ 0.002) | 0.927 ($\pm$ 0.002) |

**4.2 RuDALL-E.**  In Fig. 4 we present training dynamic of ruDALL-E[1] Malevich Ramesh et al. (2021) model on Russian Emoji dataset. The dataset Shonenkov et al. (2021) contains 2749 unique emoji icons and 1611 unique texts that were collected by web scrapping (the difference in quantities is due to the fact that there are sets, within which emojis differ only in color, moreover, some elements are homonyms in Russian). ruDALL-E Malevich is a big multimodal pretrained transformer, which learns the conditional distribution of images given some string of text (more precisely it autoregressively models the text and image tokens as a single stream of data). ruDALL-E Malevich encoder part is a 24 layer Transformer Vaswani et al. (2017) model with 16 attention heads, 2048 hidden dimensions and standard GELU nonlinearity, which in total has 1.3B parameters. It works with 128 text tokens, which are prepared from the text input using YTTM tokenizer[2], and 1024 image

---

[1]Implementation is taken from `https://github.com/sberbank-ai/ru-dalle`
[2]Implementation is taken from `https://github.com/VKCOM/YouTokenToMe`

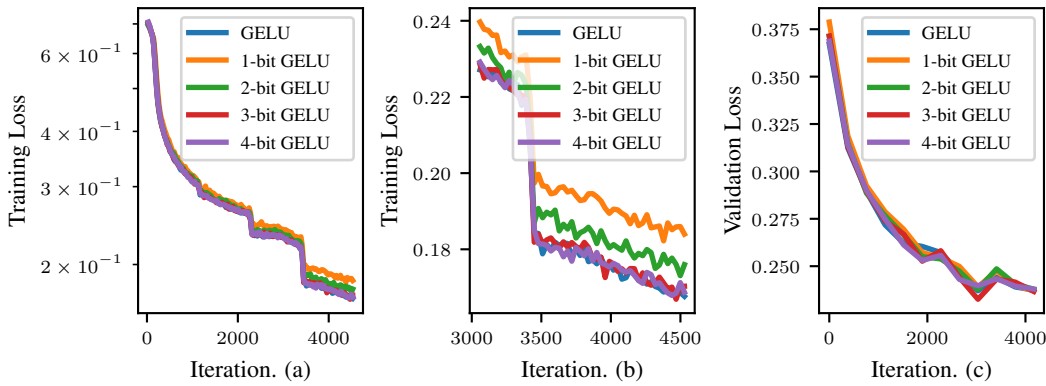

Figure 5: RoBERTa-base on QQP task from GLUE benchmark, averaged across 10 runs. (a): Training loss. (b): Training loss zoomed into the last third of the training. (c): Validation loss.

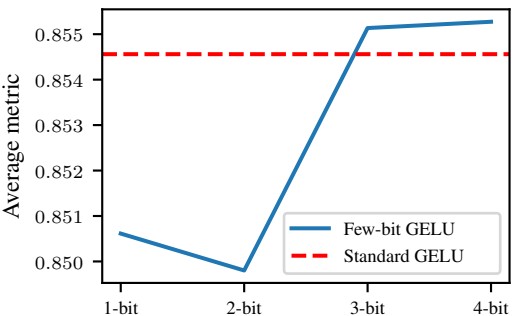

Figure 6: Task-specific metric, averaged across all tasks in GLUE benchmark. Blue line is dependence on the number of bits in Few-bit GELU and dashed red line is standard GELU. With 3 bits approximation we already match unaltered nonlinearity quality.

Figure 7: Relative top-1 accuracy for ResNet18 network on ImageNet dataset, averaged across three nonlinearities: GLUE, SELU and Swish. For each nonlinearity approximation top-1 accuracy (Few-bit approximation and ActNN approach) was measured relatively to the top-1 accuracy of the model with corresponding unaltered nonlinearity.

tokens, which are obtained after encoding the input image using Sber-VQGAN[3]. Few-bit backward for ruDALL-E Malevich shows same behaviour as for RoBERTa-base architecture: 1- and 2-bit versions, although coping with training perfectly fine, demonstrates minor performance degradation, while 3- and 4-bit versions are indistinguishable from the original GELU.

**4.3 ResNet Architecture.** We trained ResNet18 model He et al. (2016) on ImageNet Russakovsky et al. (2015) benchmark Leclerc et al. (2022) dataset with ReLU replaced with GELU, Swish and SiLU nonlinearity functions. Graphs for Swish nonlinearity can be seen in Fig. 8 and graphs for other nonlinearities can be seen in Fig. 13 in Appendix F: 1- and 2- bits have minor performance drop, while 3- and 4- bits are on par with standard nonlinearity.

**4.4 ActNN.** As a baseline, we use another quantization scheme ActNN Chen et al. (2021). It works in a much wider spectrum of situations, as it can quantize not only pointwise nonlinearity layers but also all kinds of linear layers (convolutional and dense layers), normalization layers and pooling layers. Without going deep into details, ActNN divides the saved tensor $H$ into chunks $\mathbf{h}_i$ where each chunk is of an equal size $G$. Then, given the quantization budget of $b$ bits, each chunk $\mathbf{h}_i$ is normalized: $\mathbf{u}_i = 2^b(h_i - \min\{\mathbf{h}_i\})/(\max\{\mathbf{h}_i\} - \min\{\mathbf{h}_i\})$, and its randomly quantized version is saved $\bar{\mathbf{u}}_i = \lceil \mathbf{u}_i \rceil$ with prob. $\mathbf{u} - \lfloor \mathbf{u}_i \rfloor$, $\lfloor \mathbf{u}_i \rfloor$ otherwise. Random rounding is performed in order to guarantee that the quantization is unbiased. For each group, two additional values $\min\{\mathbf{h}_i\}$ and

---

[3]Implementation is taken from `https://github.com/sberbank-ai/sber-vq-gan`

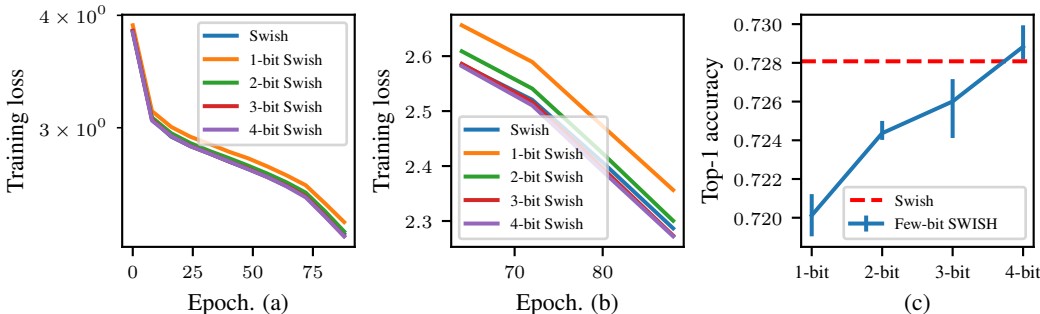

Figure 8: ResNet18 with ReLU replaced with Swish nonlinearity trained on Imagenet. (a): Training loss. (b): Training loss zoomed into the last third of the training. (c): Final validation top-1 accuracy. All graphs are averaged across three runs with different seeds. Error bars denote minimum and maximum values.

$\max\{\mathbf{h}_i\}$ are saved as well, but for the group size of $G = 256$ it is only $0.125$ additional bits per element, which we ignore in our following tests.

ActNN by construction does not take into account the global behaviour of the nonlinearity derivative. We argue that for nonlinearity layers, it is very crucial, and thus our preoptimized quantization scheme is more preferable. To confirm that, we consider ActNN behaviour on the QQP task from the GLUE benchmark with respect to different quantization budgets and compare it with our method (Fig. 9 and Table 2). In general, our method with 1 bit less budget works the same or better than ActNN, which is very important in the low-bit setting.

In Fig. 10 we compare ActNN and Few-bit for ResNet18 architecture on ImageNet dataset for SELU nonlinearity, while results for GELU and Swish nonlinearities can be found in Fig. 14 in Appendix F. Aggregated top-1 accuracy for all activation functions is presented in Fig. 7. Our method steadily outperform ActNN which is especially noticeable for 1-bit regime: ActNN experience strong downgrade of accuracy, while Few-bit Backward has much closer performance to standard nonlinearities. This means that one-bit Few-bit backward can be used in cases where it is very important to reduce memory consumption by a neural network.

|       | ActNN | Our |
|-------|-------|-----|
| 1-bit | 0.8880 ($\pm$0.0008) | **0.9080** ($\pm$0.0006) |
| 2-bit | 0.9072 ($\pm$0.0005) | **0.9097** ($\pm$0.0006) |
| 3-bit | 0.9106 ($\pm$0.0003) | **0.9114** ($\pm$0.0007) |
| 4-bit | 0.9113 ($\pm$0.0006) | **0.9114** ($\pm$0.0005) |

Table 2: Accuracy on QQP task from GLUE benchmark for ActNN and Few-bit (Our). Averaged across 5 runs. Few-bit approach is better for each memory budget. '

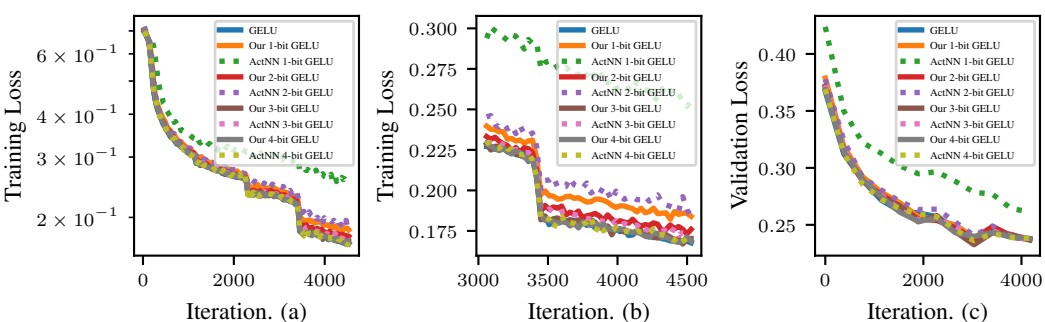

Figure 9: Comparison of RoBERTa-base on QQP task from GLUE benchmark with ActNN quantization and Few-bit approximation. Averaged across ten runs. (a): Training loss. (b): Training loss zoomed into the last third of the training. (c): Validation loss.

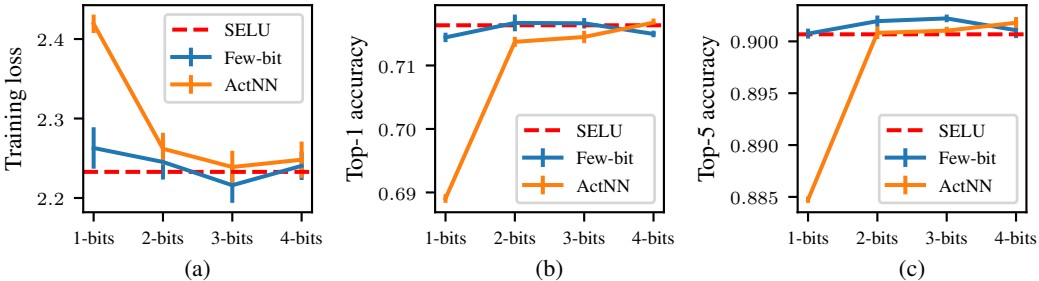

Figure 10: Comparison of ActNN SELU with Few-bit SELU (Our) for ResNet18 architecture on ImageNet dataset. (a) Training loss. (b) Top-1 accuracy. (c) Top-5 accuracy. Our method with 1-bit already matches unaltered nonlinearity performance and significantly outperform 1-bit ActNN.

## 5 RELATED WORK

The reduction of the memory footprint is an important topic. To save memory during training, in addition to working with stored activations, the memory used to store model parameters can be compressed. Quantization Bondarenko et al. (2021); Bengio et al. (2013); Banner et al. (2019); Jacob et al. (2018); Nagel et al. (2021); Krishnamoorthi (2018) limits the admissible values of weights to some small finite set. Thus, less memory is needed for storage. The low-rank representation of weights Hrinchuk et al. (2020); Phan et al. (2020); Gusak et al. (2019; 2021); Cui et al. (2020); Novikov et al. (2018); Lebedev et al. (2015) assumes some internal structure of model weights and saves memory by explicitly using this structure with low-rank methods from linear algebra. Low precision learning and low precision optimizers focus on using the lower precision floats to store weights, optimization parameters, and model gradients. All of these approaches are complementary to the proposed one and can be used together.

Checkpointing Beaumont et al. (2019; 2021); Chen et al. (2016) methods save memory by the cost of more calculations. It stores a fewer number of activations and repeats the calculation of the rest from the saved checkpoints. Offloading methods Beaumont et al. (2020) send the saved activations to the computer's RAM and load them back to the video memory on the backwards passes, which also saves GPU memory at the cost of host-device communication time.

ActNN Chen et al. (2021) is a framework for quantizing stored activations adaptively on the fly. In contrast to our work, it allows quantizing not only layers of element-by-element activations but also many others, including convolutional, normalization, and linear layers. However, this method depends on the distribution of elements of quantizable tensors and, because of that, its performance may degrade. Our approach, on the other hand, selects data-agnostic optimal quantization, which in practice turns out to be sufficient and easier to use.

## 6 CONCLUSION

We have proposed a method to reduce memory consumption during the training of deep neural network models by storing less information for backward pass in the element-wise activation functions. For effective training, there is no need to calculate the derivative of the activation functions precisely, but only its piecewise-constant approximation is sufficient. This makes it possible to save not the entire input tensor at each application of the activation function, but only the interval number in the piecewise-constant approximation. Experiments show that for a wide class of models and problems, storing only 3 bits of information per tensor element does not lead to degradation of the learning quality and saves about 20 percent of memory. We have proposed an efficient algorithm for constructing an optimal piecewise-constant approximation. The proposed drop-in replacements for popular activation functions (ReLU, GELU, Swish, Sigmoid and others) do not depend on the neural network model, the problem to be solved, or the peculiarities of data distribution. The replacement of the original activation functions by the proposed method can be performed at any training stage (both to models trained from scratch and to pre-trained models for subsequent fine-tuning) and does not require any changes in the training pipelines. An efficient CUDA implementation of the proposed method, together with pre-computed piecewise-constant approximations for many popular activation functions, is available for PyTorch at GitHub repository[4].

---

[4]Source code repository can be found at `https://github.com/anonymous/repository`

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

# A DETAILED EXAMPLES OF FEW-BIT APPROXIMATIONS FOR POPULAR NONLINEARITY LAYERS

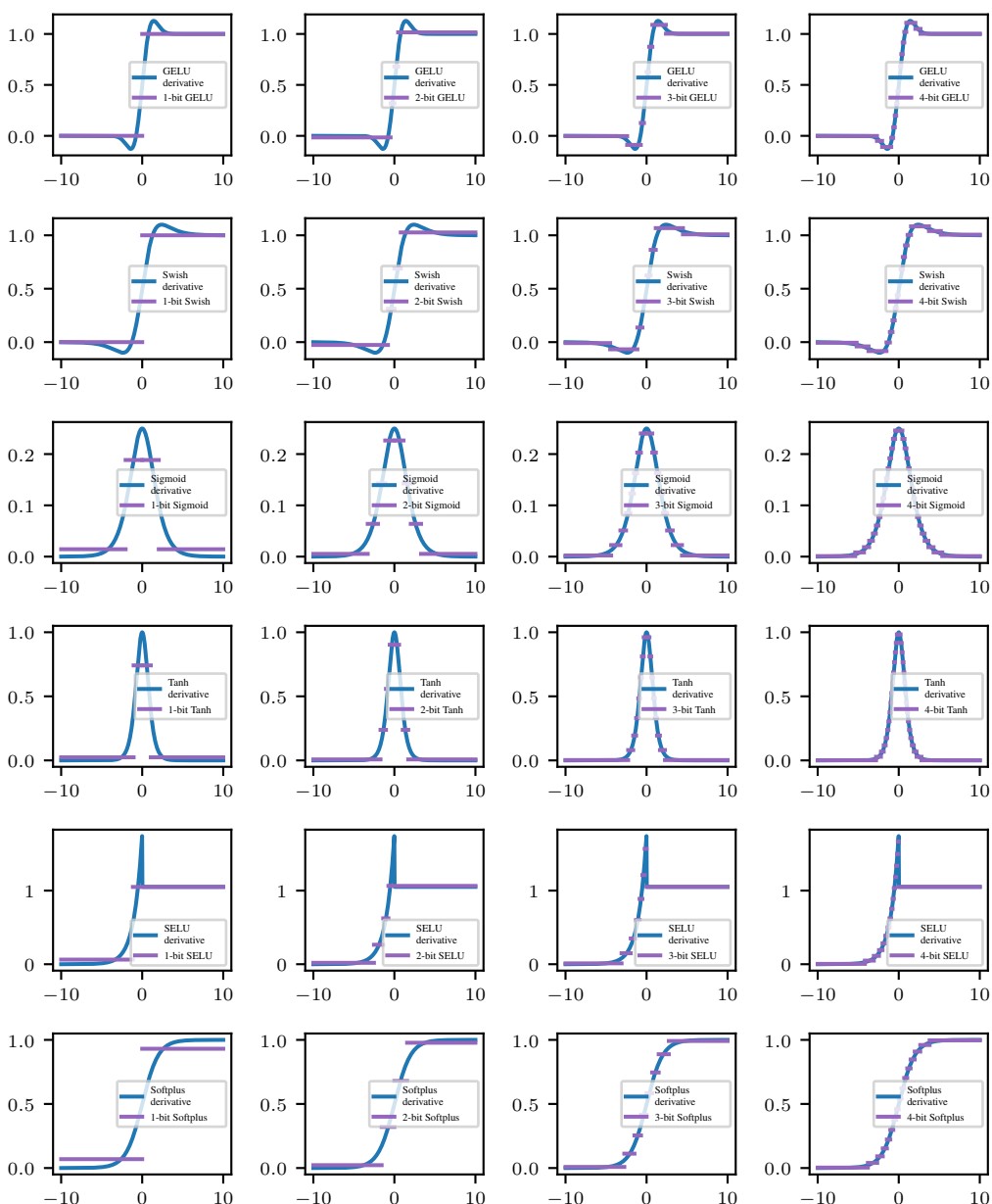

Figure 11: 1- to 4-bit approximations of popular nonlinearty layers.

# B DETAILED MEMORY MEASUREMENTS FOR DIFFERENT MODELS

We provide memory measurements for different model architectures in Table Appendix B. "Model size" is the total memory used for storing model parameters (without model gradients and optimizator statistics). "All activations size" is the total memory used by tensors, saved for backward pass. "Non-linearity activations size" is the part of all activations used only by nonlinearity layers. "Percentage saving" is memory saved on all activation using our method compared to full precision non-linearities, and percentage value in the "Maximum Batch Size" row is the increase in the batch size achievable by using our method compared to full precision non-linearities, taken in ideal circumstances. Maximum batch size is calculated with the assumption, that four model copies are stored on the device (model parameters, model gradients and optimizer statistics like two moments stored by Adam optimizer) for GPU with 32G memory.

| | Model Size (Mb) | All Act. Size (Mb) | Nonlin. Act. Size (Mb) | Standard Nonlin. Max batch size | 1-bit Max batch size | 2-bit Max batch size | 3-bit Max batch size | 4-bit Max batch size |
|---|---|---|---|---|---|---|---|---|
| **ResNet-18** | 44.6 | 40.0 | 11.5 | 813 | 1127 (+38.6%) | 1113 (+36.9%) | 1100 (+35.3%) | 1086 (+33.6%) |
| **ResNet-50** | 99.2 | 156.8 | 47.9 | 206 | 293 (+42.2%) | 289 (+40.3%) | 285 (+38.3%) | 281 (+36.4%) |
| **ResNet-101** | 171.4 | 234.5 | 73.4 | 136 | 196 (+44.1%) | 193 (+41.9%) | 190 (+39.7%) | 188 (+38.2%) |
| **ResNet-152** | 232.3 | 328.2 | 104.9 | 97 | 140 (+44.3%) | 138 (+42.3%) | 136 (+40.2%) | 134 (+38.1%) |
| **DenseNet-121** | 30.9 | 243.8 | 79.1 | 133 | 195 (+46.6%) | 192 (+44.4%) | 189 (+42.1%) | 186 (+39.8%) |
| **DenseNet-161** | 112.4 | 458.8 | 147.0 | 70 | 102 (+45.7%) | 100 (+42.9%) | 99 (+41.4%) | 97 (+38.6%) |
| **DenseNet-169** | 54.7 | 296.3 | 95.3 | 109 | 159 (+45.9%) | 157 (+44.0%) | 155 (+42.2%) | 152 (+39.4%) |
| **DenseNet-201** | 77.4 | 382.2 | 123.9 | 84 | 123 (+46.4%) | 122 (+45.2%) | 120 (+42.9%) | 118 (+40.5%) |
| **Efficient Net B0** | 20.4 | 112.4 | 32.4 | 290 | 403 (+39.0%) | 398 (+37.2%) | 393 (+35.5%) | 388 (+33.8%) |
| **Efficient Net B3** | 47.5 | 218.6 | 59.5 | 149 | 202 (+35.6%) | 200 (+34.2%) | 197 (+32.2%) | 195 (+30.9%) |
| **Efficient Net B7** | 256.3 | 674.8 | 179.3 | 47 | 63 (+34.0%) | 62 (+31.9%) | 61 (+29.8%) | 61 (+29.8%) |
| **VGG 11** | 507.2 | 100.9 | 37.0 | 304 | 472 (+55.3%) | 464 (+52.6%) | 456 (+50.0%) | 448 (+47.4%) |
| **VGG 16** | 528.2 | 163.8 | 68.5 | 187 | 314 (+67.9%) | 307 (+64.2%) | 301 (+61.0%) | 295 (+57.8%) |
| **VGG 19** | 548.4 | 178.8 | 75.0 | 171 | 288 (+68.4%) | 281 (+64.3%) | 275 (+60.8%) | 270 (+57.9%) |
| **RoBERTa-base** | 480.7 | 185.6 | 36.0 | 166 | 204 (+22.9%) | 203 (+22.3%) | 201 (+21.1%) | 200 (+20.5%) |
| **RoBERTa-large** | 1355.6 | 482.1 | 96.0 | 56 | 70 (+25.0%) | 69 (+23.2%) | 69 (+23.2%) | 68 (+21.4%) |
| **GPT2** | 491.0 | 297.1 | 146.2 | 103 | 198 (+92.2%) | 192 (+86.4%) | 187 (+81.6%) | 182 (+76.7%) |

## C    NUMERICAL RESULTS FOR DYNAMIC PROGRAMMING

|         | 1-bit  | 2-bits | 3-bits | 4-bits |
|---------|--------|--------|--------|--------|
| ReLU    | 0.0    | -      | -      | -      |
| GELU    | 0.1410 | 0.0406 | 0.0119 | 0.0031 |
| Swish   | 0.2150 | 0.0479 | 0.0170 | 0.0045 |
| Sigmoid | 0.0181 | 0.0038 | 0.0009 | 0.0002 |
| Tanh    | 0.1584 | 0.0319 | 0.0073 | 0.0017 |
| SELU    | 0.2554 | 0.1010 | 0.0184 | 0.0039 |
| Softplus| 0.2902 | 0.0541 | 0.0121 | 0.0029 |

Table 3: Numerical values of error Eq. (3) with uniform weight on interval [-10; 10].

# D EXPERIMENT SETUPS

## D.1 GLUE

Benchmark implementation is based on opensource Huggingface[5] implementation [6] and is available at `https://github.com/anonymous/repository`.

The following parameters were used:

| Task | Batch Size | Learning rate | Number of epochs | Warmup length |
|------|------------|---------------|------------------|---------------|
| Cola | 32 | 0.00002 | 10 | 320 |
| MNLI | 32 | 0.00001 | 10 | 7432 |
| MNLI-MM | 32 | 0.00001 | 10 | 7432 |
| MRPC | 16 | 0.00001 | 10 | 137 |
| QNLI | 32 | 0.00001 | 10 | 1986 |
| QQP | 32 | 0.00001 | 10 | 28318 |
| RTE | 16 | 0.00002 | 10 | 122 |
| SST2 | 32 | 0.00002 | 10 | 1256 |
| STSB | 16 | 0.00002 | 10 | 214 |

Common parameters are:

| Parameter | Value |
|-----------|-------|
| Optimizer | Adam |
| Adam $\beta_1$ | 0.9 |
| Adam $\beta_2$ | 0.98 |
| Adam $\epsilon$ | 1e-6 |
| Weight Decay | 0.1 |
| Float Precision | fp16 |

## D.2 RESNET

We use open source FFCV Leclerc et al. (2022) Imagenet benchmark[7] with ResNet18 parameters for one A100 Nvidia GPU `https://github.com/libffcv/ffcv-imagenet/blob/main/rn18_configs/rn18_88_epochs.yaml`.

## D.3 RUDALL-E

We used open source implementation that can be found at `https://github.com/sberbank-ai/ru-dalle`.

All experiments have following setup: training size 2474, valid size 275, loss image weight 1000, frozen MLP and attention layers, batch size 40, start lr 4e-7, max lr 1e-5, final lr 2e-8, warmup 0.1, 8bit-Adam Dettmers et al. (2021), weight decay 0.2, betas (0.9, 0.98), eps 1e-6, gradient checkpointing 24, trained for 6h using 1xA100.

---

[5] `huggingface.co`

[6] `https://github.com/huggingface/transformers/blob/main/examples/pytorch/text-classification/run_glue.py`

[7] `https://github.com/libffcv/ffcv-imagenet`

# E    COMBINATION OF ACTNN AND FEWBIT

ActNN method is more general and can be applied to the broader class of layers, while our method only focus on one class of layers – pointwise nonlinearities. In the cases when it is not enough and more memory saving is required it is possible to join these two methods and to use Fewbit for pointwise nonlinearities and ActNN for everything else. Such a combination should work better than pure ActNN, since Fewbit works better than ActNN for pointwise nonlinearity layers. To check this hypothesis we train ResNet18 on CIFAR10 dataset. We replace standard ReLU pointwise nonlinearity with GELU, compress all layers except GELU with 4-bit ActNN (since 2-bit ActNN is too much of a compression and model diverges) and GELU layers are compressed with either 2-bit ActNN or 2-bit Fewbit. On Fig. 12 you can see training loss and accuracy. ActNN + Fewbit for pointwise nonlinearities works slightly better than pure ActNN, as expected.

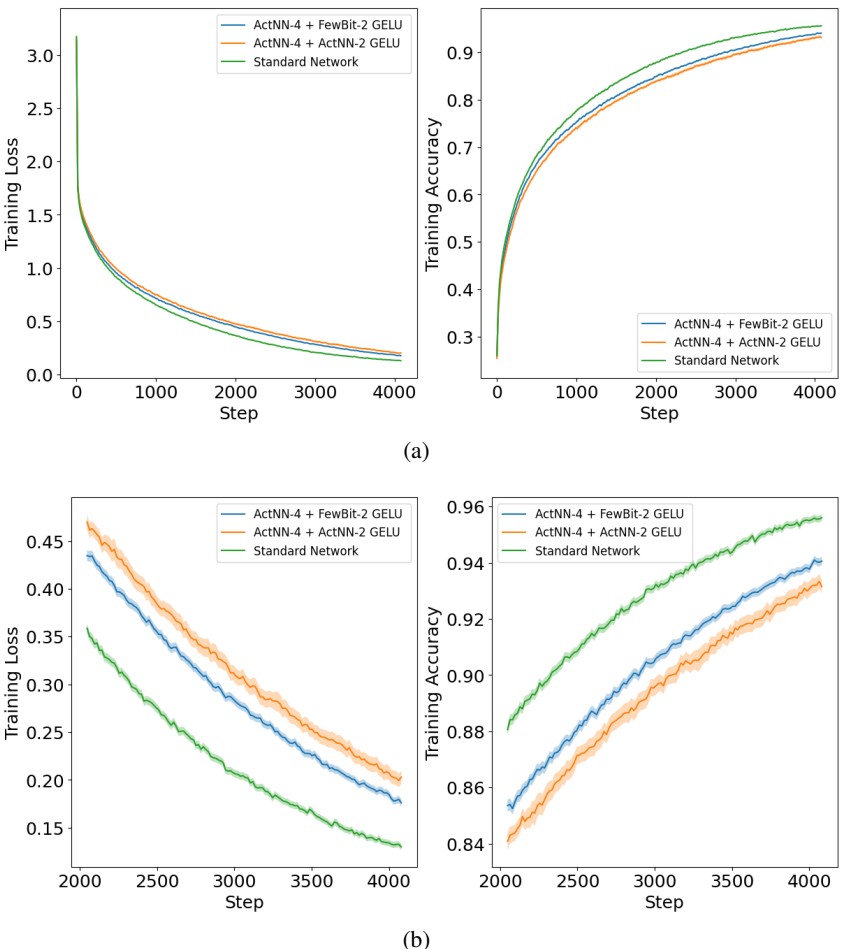

Figure 12: ResNet18 on CIFAR10 dataset. All ReLUs are replaced with GELU. All layers except pointwise nonlinearities compress their activations saved for backward with 4-bit ActNN. GELUs compress their activations saved for backward with either 2-bit ActNN (orange) or 2-bit Fewbit (blue). ResNet18 without any compresssion is depicted with green. (a): Training loss and accuracy for the whole training course. (b): Training loss and accuracy zoomed to the last half of the training course. ActNN + Fewbit for pointwise nonlinearities works slightly better than pure ActNN.

# F    MORE PLOTS FOR EXPERIMENTS

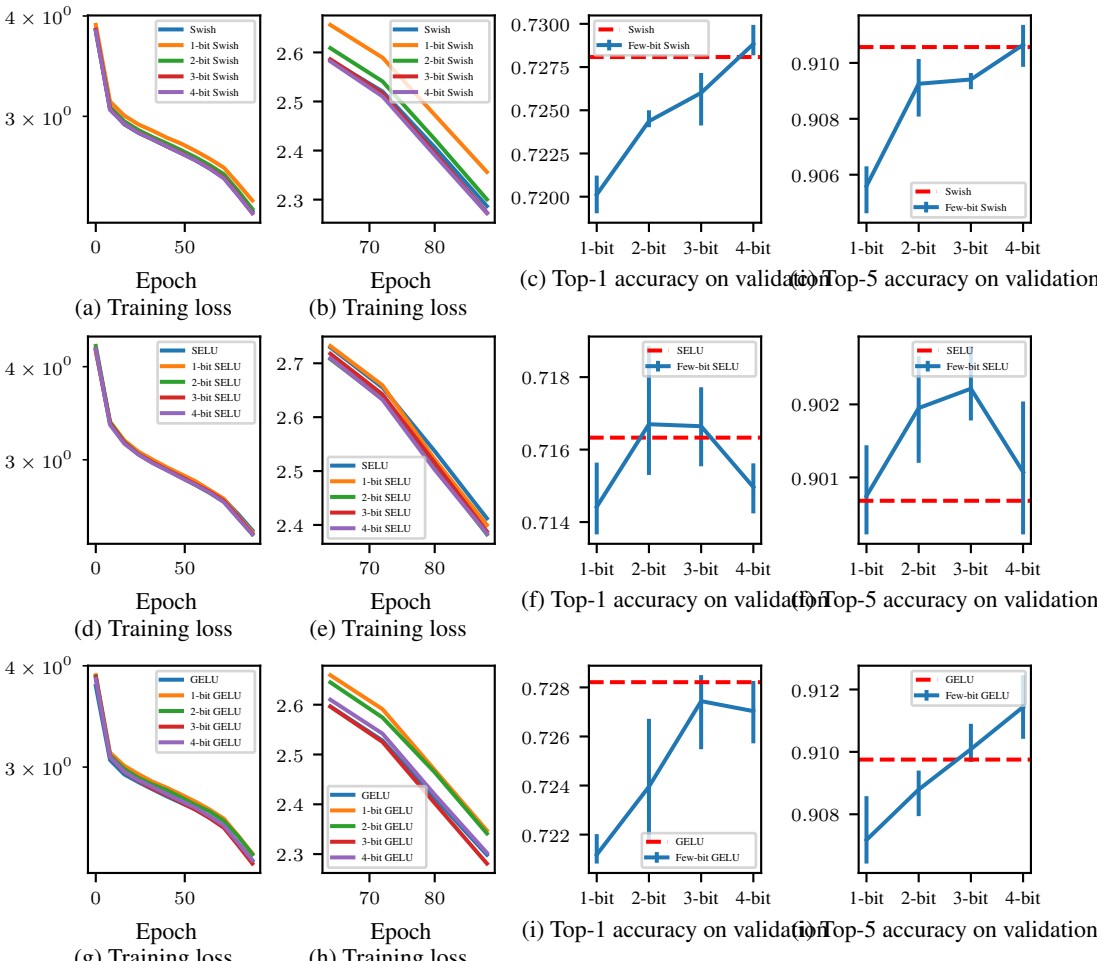

Figure 13:  ResNet18 with ReLU replaced with Swish, SELU and GELU nonlinearity trained on Imagenet. (a): Training loss. (b): Training loss zoomed into the last third of the training. (c): Final validation top-1 accuracy. All graphs are averaged across three runs with different seeds. Error bars denote minimum and maximum values.

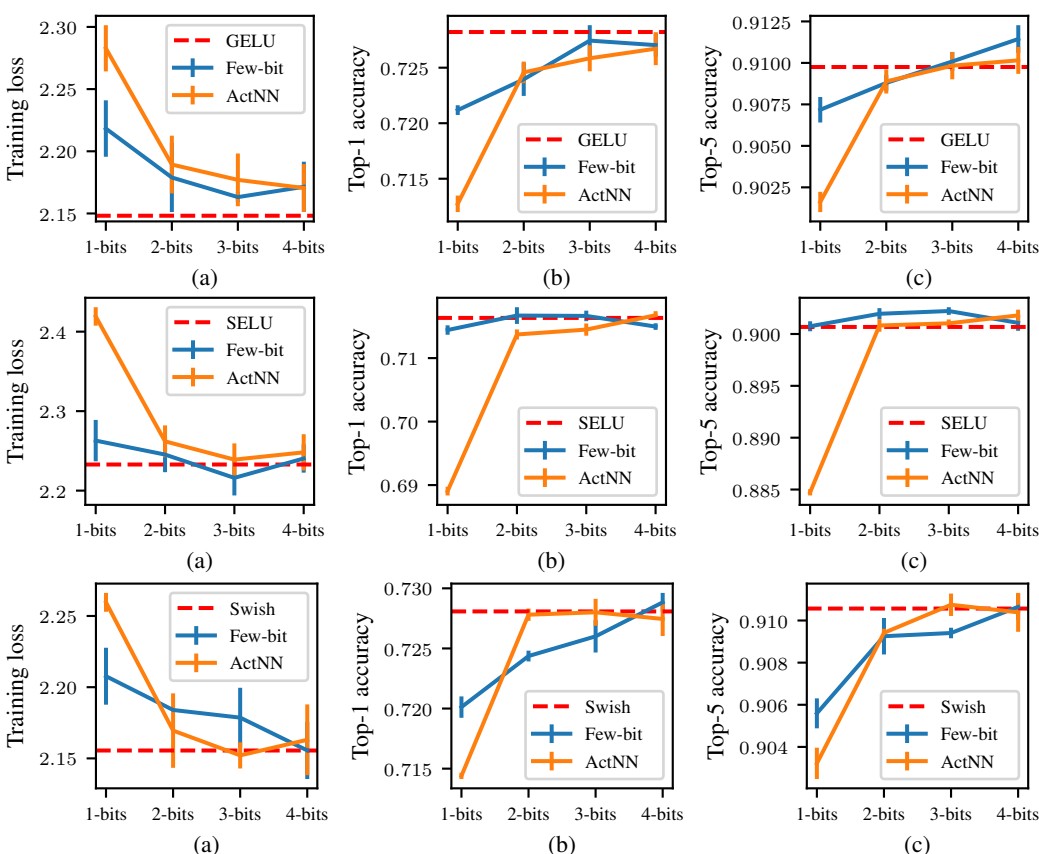

Figure 14: Comparison of ActNN GELU, SELU and Swish with Few-bit GELU, SELU and Swish (Our) for ResNet18 architecture on ImageNet dataset. (a) Training loss. (b) Top-1 accuracy. (c) Top-5 accuracy. Our method with 1-bit already matches unaltered nonlinearity performance and significantly outperform 1-bit ActNN.

## G   DYNAMIC PROGRAMMING

It is easy to see that the optimal value of $\mathbf{y}$ for $L(\mathbf{s}, \mathbf{y})$ in Eq. (3) with given $\mathbf{s}$ is:

$$y_i(\mathbf{s}) = \frac{\int_{s_i}^{s_{i+1}} w(x) f'(x) dx}{\int_{s_i}^{s_{i+1}} w(x) dx}. \tag{8}$$

Consider Eq. (7): both $y(j, i)$ and $T(j, i)$ can be calculated in advance using analytical formulas (if possible) or numerically for the corresponding 1-dimensional integrals. After that, the full array of $\mathrm{DP}(i, k)$ can be calculated in $\mathcal{O}(n^2 K)$ time and $\mathcal{O}(n^2)$ space, where $K$ is the required number of constant intervals in the approximation Eq. (2). Please note that this optimization has to be performed only once, so $n$ can be chosen quite large thus the result would be very close to the global minimum.

Note that the space complexity can be reduced to $\mathcal{O}(n)$ by adding three auxilliary arrays $F^2, W$ and $FW$ and rewriting Eq. (7):

$$
\begin{aligned}
F^2(i) &= \int_A^{t_i} f'^2(x) w(x) dx, \\
W(i) &= \int_A^{t_i} w(x) dx, \\
FW(i) &= \int_A^{t_i} f'(x) w(x) dx, \\
y(j, i) &= (FW(j) - FW(i)) / (W(j) - W(i)), \\
T(j, i) &= F^2(i) - F^2(j) - y(j, i)^2 (W(i) - W(j)).
\end{aligned}
\tag{9}
$$

We can see that ultimately only $\mathcal{O}(n)$ one-dimensional integrals have to be stored, and everything else can be easily evaluated in $\mathcal{O}(1)$ time on the spot. The one-dimensional integrals can be calculated numerically in $\mathcal{O}(n)$ time and space complexity as well:

$$
\begin{aligned}
F^2(i+1) &= F^2(i) + \int_{t_i}^{t_{i+1}} f'^2(x) w(x) dx, \\
W(i+1) &= W(i) + \int_{t_i}^{t_{i+1}} w(x) dx, \\
FW(i+1) &= FW(i) + \int_{t_i}^{t_{i+1}} f'(x) w(x) dx.
\end{aligned}
\tag{10}
$$

**Numerical results.**   In Fig. 1, we provide some 3-bit examples for popular activation functions obtained with described method, and more fewbit approximations can be seen in Fig. 11. In Table 3 we provide numerical values of error Eq. (3).

