# OpenReview forum: "Few-bit Backward: Quantized Gradients of Activation Functions for Memory Footprint Reduction"
_ICLR.cc/2023/Conference — Submitted to ICLR 2023_

### Official Review · Reviewer_1fcj · 2022-10-21

**Confidence:** 3
**Correctness:** 4
**Technical Novelty And Significance:** 2
**Empirical Novelty And Significance:** 2
**Recommendation:** 6

**Clarity, Quality, Novelty And Reproducibility:**

Clairty:
The paper is well-written. The method and the contributions are clear. Perhaps the description of the quantization algorithm could be improved. Especially equations 5-7 are could be clearer.

Quality:
The work is good quality and the reasoning is sound. The experiments could be improved as mentioned above.

Novelty:
The idea of quantizing the activations in the backward pass is novel to my knowledge.

Reproducibility:
Code is supplied along with the submission. I did not try to run the code, but I think it shouldn't be difficult to reproduce the results.

**Strength And Weaknesses:**

The algorithm for finding the quantized representation is quite clever. Its makes good use of the dynamic programming principle. I wonder, however,
* Is l2 loss the right metric to optimize? The goal is to minimize the bias that we are adding to the gradient updates and perhaps a different metric, maybe one that is optimized to minimize this bias would perform better.
* Is dynamic programming needed here? Could we simply optimize the values and the beginnings/endings of the segments using a simple optimizer?
* What is the resolution of t used in the experiments (and what interval [A, B])?

The paper makes a convincing case that the proposed method indeed reduces the memory footprint of the gradients and that this reduction does not significantly impact the training dynamics of the network. It would be nice to show concrete benefits of this memory reduction.
* As stated in the paper, the memory reduction does not result in faster backward passes. Instead, it enables training with larger batch sizes. Does training with larger batch sizes then result in a speedup?
* In the Abstract, it is stated that "Memory footprint is one of the main limiting factors for large neural network training.". Does it enable the training of larger networks that have better accuracy?
* In the conclusion, it is stated that the method results in a 20% reduction in memory usage. How is this 20% computed?
* In Appendix B, we see the potential increase in batch size. Is this table generated by measuring the size of the model running on a GPU or is it calculated according to the formula in Section 2? There may be significant differences in the theoretical memory footprints and the real memory footprints of these models.
* The paper should also compare against low-precision neural networks, for example 16-bit floating point networks. It is claimed that the savings are `complementary and can be used together', but this is not fully true because the savings don't combine multiplicatively. The savings in a low-precision network are a smaller proportion of the overall memory cost than in a high-precision network.

**Summary Of The Paper:**

The paper proposes a method for reducing the memory needs of the backward pass of a neural network. It does so by quantizing the gradients of the activation functions meaning that they now only need a few bits to store as opposed to the standard 32-bit floating point representation.

The few bit representations are computed to minimize the l2 distance from the exact gradients. The paper proposes an algorithm based on dynamic programming to compute them. This algorithm only needs to be run once for each bit count/activation function pairing.

The experiments confirm that these few-bit representation do not significantly affect the training dynamics so the memory reduction has no downside.

**Summary Of The Review:**

The paper is well written and good quality. My main issue is the thoroughness of the experiments. I would like to see the total memory reduction measured at runtime and it would also be good to see comparison against low-precision approaches (that use low-precision throughout the network not only the gradients of the non-linearities).

---

> ### Author Response · Authors · 2022-11-09
> **Is l2 loss the right metric to optimize?**
>
> > Is l2 loss the right metric to optimize? The goal is to minimize the bias that we are adding to the gradient updates and perhaps a different metric, maybe one that is optimized to minimize this bias would perform better.
>
> It is a good question. We are able to approximate the derivative with piecewise-constant functions quite well, thus the loss agnostic to dataset / neural network architecture  works better, than ActNN which is a zero-bias estimation (see Section 4.4). Since our estimation is deterministic, it has zero variance, which means that L2 loss is already a form of bias minimization. The selection of a better metric is an intriguing problem for future research, but it will require a significant amount of computational resources.

---

> ### Author Response · Authors · 2022-11-09
> **Is dynamic programming needed here?**
>
> > Is dynamic programming needed here? Could we simply optimize the values and the beginnings/endings of the segments using a simple optimizer?
>
> We tried that initially and surprisingly it does not work, independent of how hard we tried. Probably, the loss function is different from the onces encountered in neural network training. As mentioned by R2, the approach presented in the paper is similar in the logic to the famous Lloyd-Max algorithm, which is quite robust and we only need to do it once.

---

> ### Author Response · Authors · 2022-11-09
> **What is the resolution of $t$ used in the experiments (and what interval [A, B])?**
>
> > What is the resolution of $t$ used in the experiments (and what interval [A, B])?
>
> In our paper, we used the interval $[-10; 10]$ for activation functions that were discretized on $2^{13}$ intervals during dynamic programming optimization. Even with 8-bit quantization (which involves finding a piecewise-constant approximation consisting of 256 intervals), it works in a few seconds, allowing for the usage of additional discretization points (although we did not find it necessary).

---

> ### Author Response · Authors · 2022-11-09
> **Training with larger batch sizes**
>
> > As stated in the paper, the memory reduction does not result in faster backward passes. Instead, it enables training with larger batch sizes. Does training with larger batch sizes then result in a speedup?
>
> That's right, FewBits does not result in significantly faster backward passes (actually, it is a few percent faster, see the last paragraph of Section 2, but that is neglectable). Using less memory means we can incorporate larger batch. Typically, on GPUs larger batch sizes are processed more efficiently, i.e. one epoch will be faster. The main challenge is the convergence of SGD-type methods with larger batch, the topic that is quite well understood in data parallel training and solved with LAMB/LARS optimizers [1].
> We will add such discussion to the paper.
>
> [1] You, Yang, et al. "Large batch optimization for deep learning: Training bert in 76 minutes." arXiv preprint arXiv:1904.00962 (2019).

---

> ### Author Response · Authors · 2022-11-09
> **Training larger networks**
>
> > In the Abstract, it is stated that "Memory footprint is one of the main limiting factors for large neural network training.". Does it enable the training of larger networks that have better accuracy?}
>
> It surely does. It is especially useful for fine-tuning large language models, whose size has a major impact on the final quality and can be very large. Fewbit saves a significant amount of memory for it (see Table in Appendix B), and in our experiments (see Section 4.1), Fewbit completely achieves the quality of the original model with only a 3 bit memory budget for transform models. And it is remarkably simple with our efficient CUDA kernel-based Pytorch implementation: no modifications to the training pipeline, no changes to hyperparameters, no speed reduction (see the last paragraph of Section 2), thus no additional trade-off.

---

> ### Author Response · Authors · 2022-11-09
> **In the conclusion, it is stated that the method results in a 20\% reduction in memory  usage. How is this 20\% computed?**
>
> > In the conclusion, it is stated that the method results in a 20\% reduction in memory
> usage. How is this 20\% computed?
>
> Because memory reduction is heavily dependent on the network architecture and input spatial dimensions (like image size for computer vision or sequence length for language modeling), we used a minimal memory reduction rate across all of our experiments in the conclusion to be as fair as possible. 20\% memory reduction is achieved by RoBERTa-small on GLUE benchmark. For other models that were used in the experiments this value would be larger. More specific values for other architectures can be found in Table in Appendix B. This value was computed by PyTorch built-in memory profiling tools.

---

> ### Author Response · Authors · 2022-11-09
> **How Table in Appendix B was generated**
>
> > In Appendix B, we see the potential increase in batch size. Is this table generated by measuring the size of the model running on a GPU or is it calculated according to the formula in Section 2? There may be significant differences in the theoretical memory footprints and the real memory footprints of these models.
>
> We are aware of this issue, and memory measurement can be very tricky.
> Values were calculated using the formula in Section 2 and then thoroughly validated in practice by completing a full training process and monitoring real memory consumption with PyTorch builtin memory profiling tools. Theoretical values closely match practical results, and although some architectures may have minor deviations, they are negligible and have no influence on the relative comparison. We will add the discussion to the paper.

---

> ### Author Response · Authors · 2022-11-09
> **The paper should also compare against low-precision neural networks, for example 16-bit floating point networks**
>
> > The paper should also compare against low-precision neural networks, for example 16-bit floating point networks
>
> Actually, some of our experiments (namely, the one with language models) are performed in a 16-bit regime.

---

> ### Author Response · Authors · 2022-11-09
> **Fewbit and other memory-saving techniques**
>
> > It is claimed that the savings are `complementary and can be used together', but this is not fully true because the savings don't combine multiplicatively. The savings in a low-precision network are a smaller proportion of the overall memory cost than in a high-precision network.
>
> We agree on the comment about 16-bit: the savings will be less, but we have the results on training large models in a 16-bit regime and we also see the reduction in this case. We will try to highlight this point in the updated version. By "'complementary' we mean that the fewbit can be used with  other memory reduction techniques without code modification.

---

> ### Comment · Reviewer_1fcj · 2022-11-15
> **Thank you for the author reply.**
>
> Thank you for the detailed replies to the issues mentioned in the review. I am raising my score to 6. I think that the paper has merits and it is certainly interesting to practitioners.
>
> With that said, my assessment is that the paper is borderline. It could be significantly improved with a bit more polishing and experiments.
>
> Specifically, here are the changes that I recommend:
> * Appendix B should be in the main text and it should include measurements from the memory profiler.
> * The two points regarding the benefits of memory savings (speed and accuracy) should be empirically demonstrated.
> * More discussion needed regarding few-bit architectures. The evidence for the method would be more convincing it it were applied to architectures that had already been optimized for memory consumption.
> * (Point made by Reviewer Wx7m) The comparison to ActNN is a little awkward since they are doing different things. The combined result is of interest, so I am not sure that it is valid to use it as a baseline.

---

> > ### Author Response · Authors · 2022-11-18
> > **Join ActNN and Fewbit**
> >
> > Thank you very much for your reply! We are glad to hear that and will address all suggestions for improving the article.
> >
> > Regardless comparison with ActNN. Our approach and ActNN approach are solving the same problem: saving memory by compressing activations that are saved for the backward pass. ActNN method is more general and can be applied to the broader class of layers, while we focus on only one class of layers -- pointwise nonlinearities. Although it is possible to use join this to methods and to use Fewbit for pointwise nonlinearities and ActNN for everything else, it would imply much more tradeoffs during training and is practically usable for much more narrow real-life situations.
> >
> > Nevertheless, we addressed that issue and performed an experiment with ResNet18 on the CIFAR10 dataset. We used this small setup to be able to perform the necessary amount of training, but the results are quite clear. ResNet compressed with 2-bit ActNN diverges, thus we compress all layers of ResNet18 except pointwise nonlinearities with 4-bit ActNN approach and pointwise nonlinearities (GELU in that case) are compressed with either 2-bits ActNN or 2-bits Fewbit. We also demonstrate the performance of completely uncompressed ResNet18 for reference. Here are the loss (Cross-Entropy) and accuracy plots during training:
> >
> > https://ibb.co/WnNj6bX
> >
> > and the same graphs zoomed to the last third of the training process (for clarity):
> >
> > https://ibb.co/C1gv13B
> >
> > The Fewbit approach works perfectly fine in combination with ActNN and performs slightly better than ActNN for pointwise nonlinearities. For now, we added this experiment to the Appendix of the paper (Appendix E). Thank you for pointing it out!

---

### Official Review · Reviewer_KDUc · 2022-10-23

**Confidence:** 4
**Correctness:** 3
**Technical Novelty And Significance:** 2
**Empirical Novelty And Significance:** 2
**Recommendation:** 6

**Clarity, Quality, Novelty And Reproducibility:**

The method is clear. I appreciate the authors spending some time explaining the implementation details. I have some concerns about novelty that I listed above.

**Strength And Weaknesses:**

The idea is interesting however, I have a couple of concerns with the analysis.

1) The presented analysis in eq. (3) - (7) is not novel. It is simply re-deriving the famous Lloyd-Max algorithm [1].
[1] Lloyd, S. P. "Least square quantization in PCM. Bell Telephone Laboratories Paper. Published in journal much later: Lloyd, SP: Least squares quantization in PCM." IEEE Trans. Inform. Theor.(1957/1982) 18 (1957): 5.

2) How does the method compare to low-precision integer quantization, which can also be optimized (see [2])? Unlike the proposed codebook quantization scheme, low-precision integer representation can use ultra-fast tensor cores on GPUs. Can the authors provide some comparisons taking that into account?
[2] Sakr, Charbel, et al. "Optimal Clipping and Magnitude-aware Differentiation for Improved Quantization-aware Training." International Conference on Machine Learning. PMLR, 2022.

**Summary Of The Paper:**

The paper proposes to use point-wise activations to reduce the memory cost of back-propagation. An analysis is proposed to determine quantization levels minimizing a quantization MSE metric. Experimental results using GeLU and similar activations using several bitwidths are cometitve.

**Summary Of The Review:**

The presented work is an interesting approach to code-book quantization of activations. The paper is well written and has a satisfactory amount of details. However, I have some concerns regarding the novelty of the proposed quantization scheme.

---

> ### Author Response · Authors · 2022-11-09
> **Lloyd-Max algorithm**
>
> >The presented analysis in eq. (3) - (7) is not novel. It is simply re-deriving the famous Lloyd-Max algorithm [1]. [1] Lloyd, S. P. "Least square quantization in PCM. Bell Telephone Laboratories Paper. Published in journal much later: Lloyd, SP: Least squares quantization in PCM." IEEE Trans. Inform. Theor.(1957/1982) 18 (1957): 5.
>
> Thank you for the reference. We checked formulas (13) and (14) in the manuscript, i.e.
>
> $$
>     N = \sum_{\alpha = 1}^\nu \int_{Q_\alpha} (q_\alpha - x)^2 dF(x).
> $$
>
> $$
>     q_\alpha = \frac{\int_{Q_\alpha} x dF(x) }{\int_{Q_\alpha} dF(x)}, \quad \alpha = 1,2,...,\nu.
> $$
>
> are closely related to our Equations (4) and (6). But the actual Lloyd-Max iterative procedure
> $$
>     q_{\alpha}^i = \textrm{c.m. of } Q_{\alpha}^{i - 1}
> $$
> $$
>     x_{\alpha}^{i + 1} = \frac{1}{2} (q_{\alpha}^i + q_{\alpha + 1}^i)
> $$
> can not be applied in our case of nonmonotonic functions, as far, as we understand. Thus we use different approach, namely dynamic programming. Nevertheless, dynamic programming is a rather popular method to solve similar problems, so it's not the novelty of the paper, but the idea to apply a static quantization (to be done only once) for the pointwise nonlinearity derivative.

---

> ### Author Response · Authors · 2022-11-09
> **Comparison with OCTAV**
>
> > How does the method compare to low-precision integer quantization, which can also be optimized (see [2])? Unlike the proposed codebook quantization scheme, low-precision integer representation can use ultra-fast tensor cores on GPUs. Can the authors provide some comparisons taking that into account? [2] Sakr, Charbel, et al. "Optimal Clipping and Magnitude-aware Differentiation for Improved Quantization-aware Training." International Conference on Machine Learning. PMLR, 2022.
>
> The fair comparison is quite tricky because the goals of the methods are different. We were not able to find an implementation of the method, maybe you have the reference for the code (since efficient implementation using TensorCores could be quite tricky).
> From our understanding, OCTAV (formula (2) in the paper) still requires the computation of the derivative in floating point, which is quite slow, so we don't expect it would be faster than Fewbit (because Fewbit is faster than standard pointwise nonlinearity forward-backward pass, see the last paragraph of Section 2). We will add the reference to OCTAV paper, thank you!

---

### Official Review · Reviewer_Wx7m · 2022-10-24

**Confidence:** 5
**Correctness:** 3
**Technical Novelty And Significance:** 2
**Empirical Novelty And Significance:** 2
**Recommendation:** 3

**Clarity, Quality, Novelty And Reproducibility:**

The paper is mostly clearly written and reproducible. Novelty might be somewhat thin.

**Strength And Weaknesses:**

Strength:
- Reducing training memory footprint is an important problem.
- The proposed method technically sounds, and should be the correct way to deal with nonlinearities.

Weaknesses:
- The paper still needs polishing. For example, the introduction is too brief.
- The improvement is incremental.
- It is somewhat unclear whether the comparison with ActNN is fair. ActNN compresses both the linear and nonlinear layers. Does the proposed method also do so? (i.e., combine the proposed method for nonlinear layer and ActNN for linear layer. I think the combined result is what of practical interest.)
- Table lookup can be very expensive. The time consumption is not reported. I'm not sure if the proposed nonlinear quantization is practical.

**Summary Of The Paper:**

This paper proposes a method to perform activation compressed training (ACT) for pointwise non-linear activation functions. Instead of storing a quantized version of the input x, the proposed approach stores a quantized version of the activation function's gradient f'(x), which is multiplied with the gradient in the backward phase. The paper proposes a nonlinear quantization method for f'(x) with a dynamic programming method to determine the optimal quantized approximation. Quantization can be reduced to table lookup then. The proposed method is evaluated on language model pretraining, text-to-image generation, and image classification tasks, while the proposed method is slightly better than existing ACT methods under the same bitwidth.

**Summary Of The Review:**

The paper deals with an important problem in a reasonable manner. However, the paper might require more work to fully demonstrate its significance before publication.

---

> ### Author Response · Authors · 2022-11-09
> **Introduction is too brief**
>
> > The paper still needs polishing. For example, the introduction is too brief.
>
> We follow the recommendation that the introduction should not take more than one page.

---

> ### Author Response · Authors · 2022-11-09
> **The improvement is incremental.**
>
> > The improvement is incremental.
>
> The novelty of our approach is to apply static quantization to the gradients of activation functions. To our knowledge, this simple idea has not been used before and leads to significant improvements: reduction in memory without any accuracy or speed loss.

---

> ### Author Response · Authors · 2022-11-09
> **Fairness of comparison with ActNN**
>
> > It is somewhat unclear whether the comparison with ActNN is fair. ActNN compresses both the linear and nonlinear layers. Does the proposed method also do so? (i.e., combine the proposed method for nonlinear layer and ActNN for linear layer. I think the combined result is what of practical interest.)
>
> We think the comparison is fair, since in the reported results we used ActNN to compress only nonlinear layers. We agree that that the combined result is of practical interest, but requires additional work for selection of hyperparameters of ActNN.

---

> > ### Author Response · Authors · 2022-11-18
> > **Join ActNN and Fewbit**
> >
> > We performed an experiment with ResNet18 on the CIFAR10 dataset. We used this small setup to be able to perform the necessary amount of training, but the results are quite clear. ResNet compressed with 2-bit ActNN diverges, thus we compress all layers of ResNet18 except pointwise nonlinearities with 4-bit ActNN approach and pointwise nonlinearities (GELU in that case) are compressed with either 2-bits ActNN or 2-bits Fewbit. We also demonstrate the performance of completely uncompressed ResNet18 for reference. Here are the loss (Cross-Entropy) and accuracy plots during training:
> >
> > https://ibb.co/WnNj6bX
> >
> > and the same graphs zoomed to the last third of the training process (for clarity):
> >
> > https://ibb.co/C1gv13B
> >
> > The Fewbit approach works perfectly fine in combination with ActNN and performs slightly better than ActNN for pointwise nonlinearities. For now, we added this experiment to the Appendix of the paper. Thank you for pointing it out!

---

> ### Author Response · Authors · 2022-11-09
> **Time consumption**
>
> > Table lookup can be very expensive. The time consumption is not reported. I'm not sure if the proposed nonlinear quantization is practical.
>
> We tried to come up with a fast implementation and implemented FewBit backwards as a PyTorch framework package utilizing CUDA kernels. Our benchmarks show that the backward pass is faster, than the standard implementation. For example, the average time required for forward-backward pass for GPT2 model on V100 GPU has reduced from 228 ms to 217 ms (note, that this for the same batch size, i.e. the difference is only in the evaluation of the backward pass for nonlinear layers). The reason in this case that original layer requires significant number of floating point operations for the evaluation of GeLU function (it includes exp/erf/tanh). Thus, Fewbit not only reduces the memory consumption, but does not increase the computational complexity, i.e. it can be used as a drop-in replacement of the standard non-linearity functions.

---

> ### Comment · Reviewer_Wx7m · 2022-12-07
> **Thanks for the responses**
>
> Thanks for the responses and the additional experiments. However, I think thorough studies (such as the accuracy result for nonlinear + linear compression) should be included in the paper before publication. Though the authors add some result on CIFAR-10, the dataset is still small and not entirely representative. Therefore, I still think this paper cannot be published in its current form.

---

### Decision · Program_Chairs · 2023-01-20

**Decision:**

Reject

**Justification For Why Not Higher Score:**

One reviewer was negative [3] and took part in the later discussion, while two reviewers were marginally positive [6,6], and did not respond during the late discussion phase. Overall, I think the idea is nice, and the work done here is significant, but I think the paper needs more polishing before publication (fixing the weaknesses mentioned above).

**Justification For Why Not Lower Score:**

N/A

**Metareview: Summary, Strengths And Weaknesses:**

This paper suggests a method to compress the activations gradients using a piece-wise constant approximation (fitted by first using dynamic programming to find a good initialization and then using gradient descent). This saves memory during neural network training.

Strengths:
1) The proposed method is novel and non-trivial.
2) The paper is mostly clear, though parts of it needs to be polished*.
3) The code is provided and is well documented**.
4) The method was implemented using cuda kernel, which is not trivial.

Weaknesses:
1) The comparisons with the previous State-of-the-art (ACTNN) are partial, i.e. the baseline is shown only for some results. For example, I would like to see Table 1 for ACTNN, not just for the QQP task as in Table 2, to avoid any suspicion of cherry-picking.
2) The differences vs ACTNN only seem significant for 1-bit, and even there are quite small.
3) Considering that there are still other types of memory consumptions (data, model, momentum, buffers, ...), the overall memory saving of this work might be quite small (less than 10%). In comparison, previous works (e.g., BLPA, TinyScript, ActNN and GACT) also compressed the linear layers, which makes the memory saving more significant.


*A reviewer commented the paper is too colloquial for an academic publication (e.g. the first paragraph of Sec. 4).

** Note the code link in the submission is broken, so I asked a researcher to verify the quality of the code in the arxiv version (without telling me who are the authors).